# Exogenous diethyl aminoethyl hexanoate ameliorates low temperature stress by improving nitrogen metabolism in maize seedlings

**Jianguo Zhang[1,2], Shujun Li[2], Quan Cai[2], Zhenhua Wang**👤[1]*, **Jingsheng Cao[2], Tao Yu[2], Tenglong Xie[1]**

**1** College of Agriculture, Northeast Agricultural University, Harbin, P.R. China, **2** Maize Research Institute, Heilongjiang Academy of Agricultural Sciences, Harbin, P.R. China

* zhenhuawang_2006@163.com

**Data Availability Statement:** All relevant data are within the manuscript and its Supporting Information files.

## Abstract

Spring maize sowing occurs during a period of low temperature (LT) in Northeast China, and the LT suppresses nitrogen (N) metabolism and photosynthesis, further reducing dry matter accumulation. Diethyl aminoethyl hexanoate (DA-6) improves N metabolism; hence, we studied the effects of DA-6 on maize seedlings under LT conditions. The shoot and root fresh weight and dry weight decreased by 17.70%~20.82% in the LT treatment, and decreased by 5.81%~13.57% in the LT + DA-6 treatment on the 7th day, respectively. Exogenous DA-6 suppressed the increases in ammonium ($NH_4^+$) content and glutamate dehydrogenase (GDH) activity, and suppressed the decreases in nitrate ($NO_3^-$) and nitrite ($NO_2^-$) contents, and activities of nitrate reductase (NR), nitrite reductase (NiR), glutamine synthetase (GS), glutamate synthase (GOGAT) and transaminase activities. NiR activity was most affected by DA-6 under LT conditions. Additionally, exogenous DA-6 suppressed the net photosynthetic rate (Pn) decrease, and the suppressed the increases of superoxide anion radical ($O_2^{\cdot-}$) generation rate and hydrogen peroxide ($H_2O_2$) content. Taken together, our results suggest that exogenous DA-6 mitigated the repressive effects of LT on N metabolism by improving photosynthesis and modulating oxygen metabolism, and subsequently enhanced the LT tolerance of maize seedlings.

## Introduction

Throughout the growing season, crops frequently suffer from various types of environmental stress. As one of the major abiotic stresses, low temperature (LT) negatively affects plant growth and development [1]. Brief exposure to LT may disrupt plant physiological processes, such as water status, photosynthesis and nitrogen (N) metabolism, but plants generally survive [2,3]; prolonged exposure to LT may lead to plant necrosis or death. Northeast China is one of the major agricultural production areas in China, accounting for approximately 20% of total domestic grain production. This region has a typical temperate continental monsoon climate

**Funding:** The National Key Research and Development Program of China (2017YFD0101101), and the Special Construction of Modern Agricultural Industry Technology System(CARS-02-05).

**Competing interests:** The authors have declared that no competing interests exist.

with few heat resources. The frost-free period of the whole year is generally 100–150 days, and plants frequently encounter LT during the spring sowing stage and seedling growth stage, which negatively affects agricultural production [4].

As the crop with the third largest cultivated and highest yield worldwide, maize (*Zea mays* L.) plays an important role in ensuring global food security [5]. As a thermophilic $C_4$ plant that originates from subtropical regions, maize growth is highly susceptible to LT. N metabolism, including N uptake, transport, reduction and assimilation as well as amino acid metabolism, is a fundamental process in plants [6]. Moreover, most plant stress-responsive physiological processes involve N metabolism, such as enhanced nutrient uptake and transport, improved photosynthetic regulation, rapid synthesis of osmotic solutes and structural alterations [7]. Hence, N metabolism is extremely important for the growth and LT tolerance of plants.

Chemical regulation is widely applied in agricultural production as a strategy to prevent or alleviate the adverse effects induced by abiotic stresses. Diethyl aminoethyl hexanoate (DA-6), a plant growth regulator, is involved in the regulation of a wide range of metabolic and physiological responses of crop plants such as maize, cotton, soybean, peanut, tomato and wheat [8–11]. Exogenous DA-6 increases grain weight through involvement in the synthesis of sucrose and starch [8]; promotes seeds germination and seedling establishment by mediating fatty acid metabolism and glycometabolism [9]; enhances seedling growth through altered photosynthesis by accelerating chlorophyll biosynthesis and increasing the activities of phosphoenolpyruvate carboxylase (PEPcase) and ribulose-1,5-bisphosphate carboxylase (RuBPcase); and regulates hormone balance by enhancing the contents of auxin, zeatin riboside and gibberellin but decreasing the content of abscisic acid [10]. Exogenous DA-6 also has positive effects on the improvement of plant stress resistance, such as resistance to salinity stress and heavy metal stress [11,12].

Despite accumulating research that has enriched our understanding of the improvement of growth and development following DA-6 application, the possible role of DA-6 in alleviating LT stress has not yet been explored. In this study, N metabolism, photosynthesis and the antioxidant system were examined to investigate whether exogenous DA-6 could enhance the LT resistance of maize seedlings and how doses exogenous DA-6 affect N metabolism in stressed plants.

## Materials and methods

### Material and growth conditions

Maize (*Zea mays* L.) inbred line Q319 and DA-6 were obtained from the Heilongjiang Academy of Agricultural Sciences and the China Zhengzhou Zhengshi Chemical Limited Company, respectively. After sterilization (0.2% $HgCl_2$ for 10 min and rinsing with abundant distilled water), seeds were soaked in deionized water for 24 h and then germinated in Petri dishes at 28˚C for 96 h in the dark. Afterward, uniformly germinated seedlings were transferred to opaque plastic containers containing 20 L of 1/2 modified Hoagland's nutrient solution, which was continuously aerated and adjusted to 6.30 (±0.05) daily. The whole experiment was conducted in a controlled growth room under the following conditions: relative humidity 60–70%, light intensity 350 μmol·m$^{-2}$·s$^{-1}$ and a 15-h photoperiod.

### Experimental design and sampling

In our preliminary experiment, a wide range of temperatures (9˚C~15˚C) and various concentrations of DA-6 (5, 10, 15, 20 and 25 mg/L) were employed. Finally, 11˚C and 10 mg/L DA-6 were chosen on the basis of the growth parameters as the optimum combination for investigating the effects of DA-6 on maize seedlings. Seedlings at the three-leaf stage were exposed to

treatments in different nutrient solutions as follows: (1) Control = nutrient solution was not supplemented with DA-6 under non-LT conditions (28±1˚C); (2) DA-6 = nutrient solution was supplemented with DA-6 under non-LT conditions (28±1˚C); (3) LT = nutrient solution was not supplemented DA-6 under LT conditions (11±1˚C); and (4) LT + DA-6 = nutrient solution was supplemented with DA-6 under LT conditions (11±1˚C). There were a total of 100 plants per container, and the nutrient solution was aerated daily at 7:00~9:00, 11:00~13:00 and 15:00~17:00.

For growth parameters and root hydraulic conductivity (Lp) measurements, plants were sampled only on the 7th day after LT stress. At 0, 1, 3, 5 and 7 days after LT treatment, the 3rd leaves from the base of the seedlings were sampled for gas exchange parameters. A total of 15 plants were sampled at each sampling time from each container. The leaves were immediately frozen in liquid N, stored at −80˚C and used for related analyses.

## Plant measurements and analysis

### Growth parameters and Lp

The fresh weight (FW) of roots and shoots was measured after the plants were harvested and immediately divided. After FW measurement, the plants were oven-dried at 105˚C for 30 min and held at 80˚C for 48 h to obtain dry weight (DW). The mean values of 10 plants were considered one replication. The length, surface area and volume of roots were measured using the WinRHIZO Image Analysis system (Version 2013e) (Regent Instruments Inc., Canada). The Lp was assayed with a Scholander pressure chamber according to the description of López-Pérez et al. (2007) [13].

### RNA isolation and real-time RT-PCR

Total RNA was extracted from the maize roots using TRIzol reagent (Invitrogen, Carlsbad, CA, USA). The gene-specific primers are listed in S1 Table. The synthesis of cDNA and real-time PCR were performed as previously described by Liu et al. (2012) [14]. The relative expression of the target genes was calculated using the $2^{-\triangle\triangle Ct}$ method [15].

### Related indicators of photosynthesis, antioxidant system and N metabolism

The gas exchange parameters of seedlings at the 3~4 leaf stage were assayed with a calibrated portable LI-6400 gas exchange system (*Li-6400*, *Li-Cor Inc.*, USA) that maintained an external $CO_2$ concentration at $380 \pm 10$ μmol mol$^{-1}$ and a light intensity of 1,000 lmol photons·m$^{-2}$·s$^{-1}$. The 3th leaf (numbered basipetally) was sampled, and the measurements were performed from 10:00~12:00.

The total chlorophyll content was measured based on the chlorophyll absorbances by the supernatant measured at 663 nm according to the method of Arnon (1949) [16]. The activities of PEPcase and RuBPcase were assayed according to Omoto et al. (2012) [17] and Xie et al. (2017) [18], respectively.

The generation rate of superoxide anion radicals ($O_2\cdot^-$) and hydrogen peroxide ($H_2O_2$) content were determined according to the methods of Elstner and Heupel (1976) [19] and Jana and Choudhuri (1982) [20], respectively.

Superoxide dismutase (SOD) activity was determined by measuring its ability to inhibit the photochemical reduction of NBT as described by Giannopolitis and Ries (1977) [21]. Peroxidase (POD) activity was measured according to the guaiacol method described by Zheng and Huystee (1992) [22]. Catalase (CAT) activity was measured as described by Aebi (1984) [23].

APX activity (EC 1.11.1.11) was measured by monitoring the decrease in AsA absorbance at 290 nm according to the guaiacol method described by Nakano and Asada (1980) [24].

The contents of foliar $NO_3^-$, $NO_2^-$ and $NH_4^+$ were determined according to the methods of Cataldo et al. (1975) [25], Barro et al. (1991) [26] and Bräutigam et al. (2007) [27].

The activities of foliar nitrate reductase (NR), nitrite reductase (NiR) and glutamine synthase (GS) were determined as described by Barro et al. [28], Ida and Morita [29] and O'neal and Joy [30], respectively. The activities of foliar glutamine oxoglutarate aminotransferase (GOGAT) and glutamate dehydrogenase (GDH) were determined as described by Groat and Vance [31]. The activities of foliar alanine aminotransferase (AlaAT) and aspartate aminotransferase (AspAT) were determined according to the methods of Jia et al. (2015) [32].

## Free amino acid and soluble protein contents and proteinase activity

The contents of free amino acids and soluble protein, and the protease activity were determined by the methods of Yemm and Cocking (1955) [33], Bradford (1976) [34] and Drapeau (1974) [35], respectively.

## Statistical analysis

The experiment used a randomized complete block design (RCBD), and 5 experimental replications were considered during statistical analysis. The data were analysed using the Software Package for Social Science (SPSS) version 17.0, and all of the values are presented as the mean ± SE. Tukey's test at the 5% probability level was applied to examine the differences among mean values on a given day of stress treatment. The results are indicated in tables and figures such that the letters a, b c, and d represent the first, second, third, and fourth levels of statistical significance, respectively.

## Results

### Effects of LT and/or exogenous DA-6 on growth parameters

Exogenous DA-6 promoted growth under non-LT conditions and partially alleviated the growth inhibition induced by LT (Table 1). Compared with those in the control, shoot FW, root FW, shoot DW and root DW decreased by 18.75%, 19.92%, 20.82% and 17.70% in the LT treatment, decreased by 7.04%, 5.81%, 13.57% and 9.06% in the LT + DA-6 treatment, and increased by 6.83%, 7.92%, 6.43% and 7.51% in the DA-6 treatment, respectively.

The values represent the mean±SE (n = 5). Values with the same letters in the columns are not significantly different at P<0.05 (Tukey test). Control: Non-low temperature conditions (28±1˚C), DA-6: Diethyl aminoethyl hexanoate treatment under non-low temperature conditions (28±1˚C), LT: Low temperature conditions (11±1˚C), LT + DA-6: Diethyl aminoethyl hexanoate treatment under low temperature conditions (11±1˚C).

**Table 1. Effects of LT and/or DA-6 treatment on the fresh weight (FW) and dry weight (DW) of the shoots and roots of the maize seedlings on the 7[th] day after LT stress (4-leaf stage).**

| Treatment | FW (g·plant⁻¹) | | DW (g·plant⁻¹) | |
|---|---|---|---|---|
| | Shoot | Root | Shoot | Root |
| **Control** | 1.948±0.074ab | 0.847±0.023b | 0.156±0.005a | 0.072±0.001a |
| **DA-6** | 2.081±0.076a | 0.914±0.049a | 0.166±0.003a | 0.078±0.004a |
| **LT** | 1.583±0.086c | 0.678±0.014c | 0.124±0.007c | 0.059±0.006b |
| **LT+DA-6** | 1.811±0.096b | 0.798±0.029b | 0.135±0.007b | 0.066±0.002b |

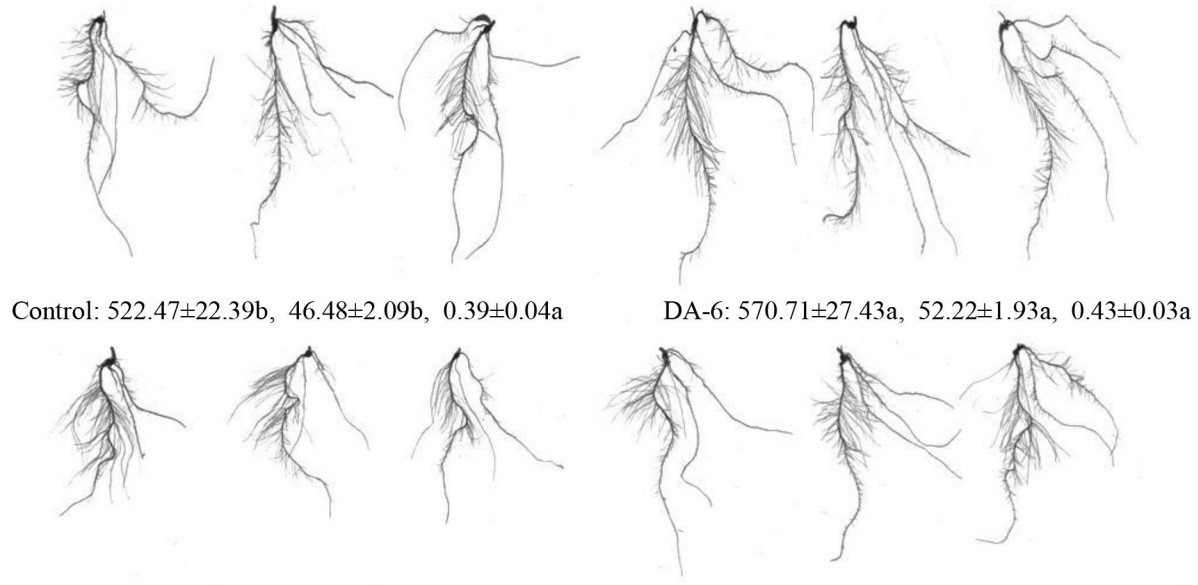

Control: 522.47±22.39b, 46.48±2.09b, 0.39±0.04a        DA-6: 570.71±27.43a, 52.22±1.93a, 0.43±0.03a

LT: 277.72±23.93d, 16.96±2.48d, 0.16±0.02c        LT+DA-6: 365.56±18.28c, 26.88±2.58c, 0.26±0.06b

**Fig 1. Effects of LT and/or exogenous DA-6 treatment on the root morphology of maize seedlings on the 7th day.** Data in the figure are treatment: root length (cm), surface (cm$^2$) and volume (cm$^3$), in order. Values with the same letters are not significantly different at P<0.05 (Tukey test).

Exogenous DA-6 positively impacted the root morphology of plants under non-LT and LT conditions (Fig 1). Compared with those in the control, the length, surface and volume of roots decreased by 46.85%, 63.50% and 59.95% in the LT treatment, decreased by 30.03%, 42.17% and 33.01% in the LT + DA-6 treatment, and increased by 9.23%, 12.33% and 10.04% in the DA-6 treatment, respectively.

## Effects of LT and/or exogenous DA-6 on Lp

Exogenous DA-6 partially suppressed the decrease in Lp during LT (Fig 2). On the 1st, 3rd, 5th and 7th days, the Lp decreased by 41.23%, 44.59%, 53.92% and 53.51% in the LT treatment;

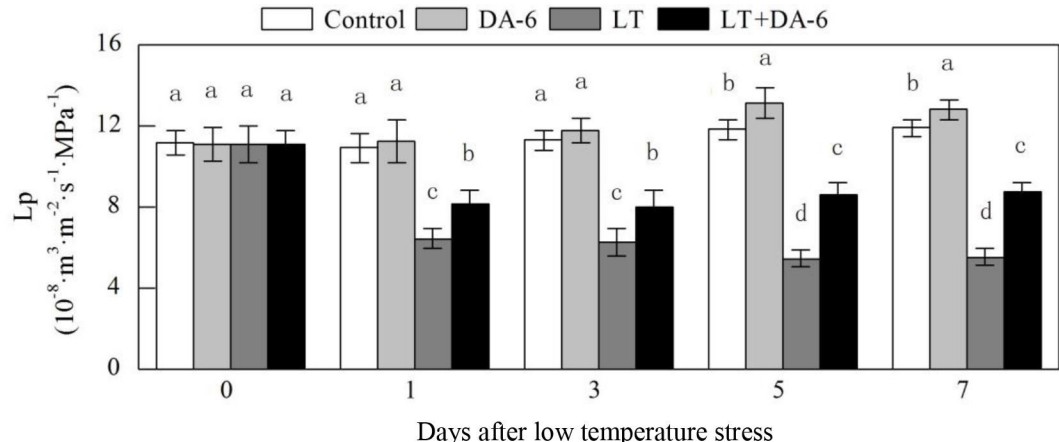

**Fig 2. Effects of LT and/or exogenous DA-6 on Lp.** The values represent the mean ± SE (n = 5). Values with the same letters in the columns are not significantly different at P<0.05 (Tukey test).

decreased by 33.31%, 36.24%, 44.21% and 43.91% in the LT+DA-6 treatment; and increased by 25.13%, 29.26%, 27.60% and 26.61% in the DA-6 treatment, respectively, compared with the levels in the control. Exogenous DA-6 significantly increased Lp on the 5th and 7th days by 10.39% and 7.53%, respectively, compared with the levels in the control.

## Effects of LT and/or exogenous DA-6 on the relative expression levels of NRT 1;1, NRT 1;2 and NRT 2;5

Exogenous DA-6 suppressed the downregulated relative expression levels of NRT 1;1, NRT 1;2 and NRT 2;5 during LT (Fig 3). On the 1st, 3rd, 5th and 7th days, NRT 1;1 relative expression

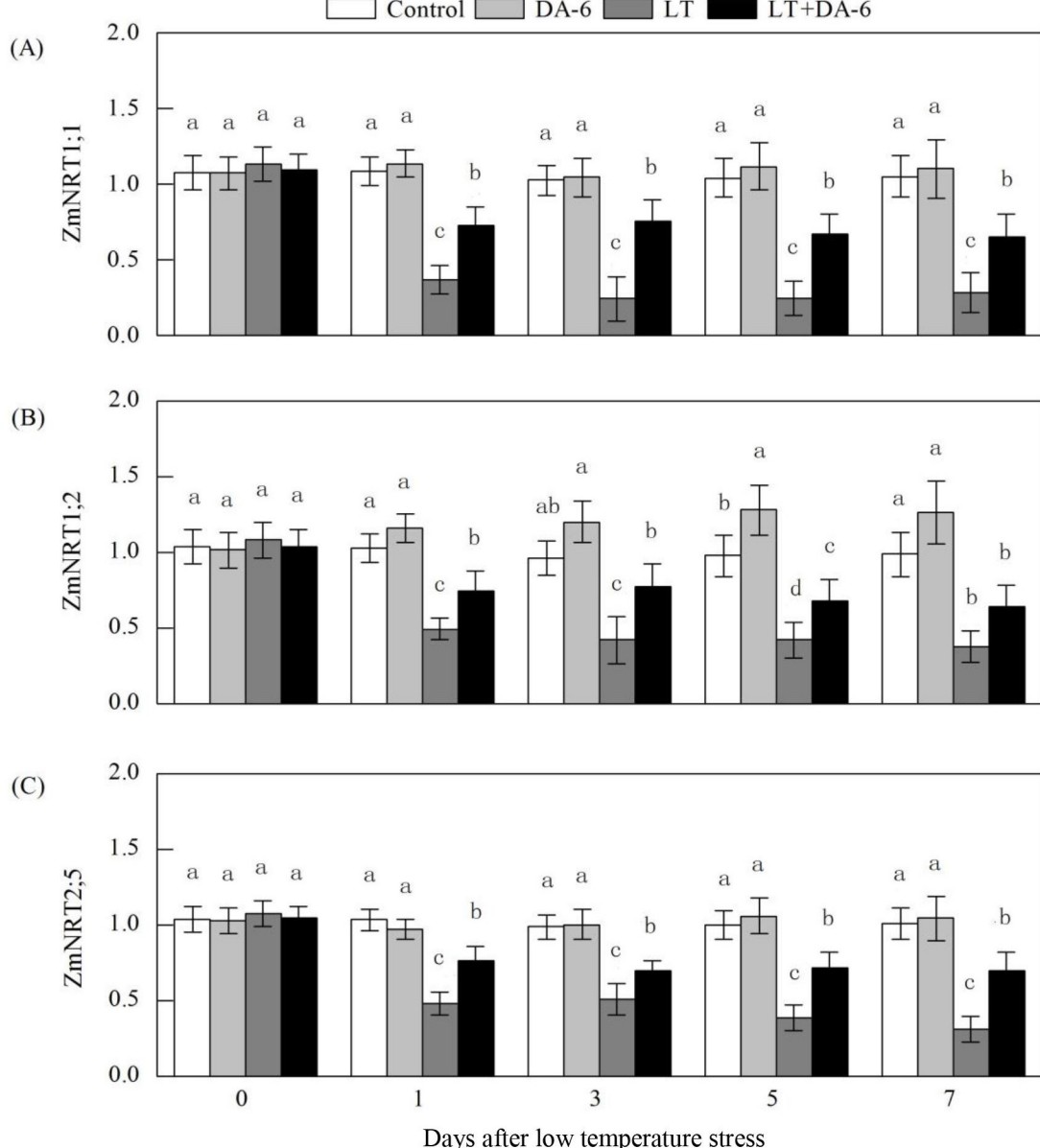

**Fig 3.** Effects of LT and/or exogenous DA-6 on the relative expression levels of NRT1;1 (A), NRT1;2 (B) and NRT2;5 (C). The values represent the mean ± SE (n = 5), and values with the same letters in the columns are not significantly different at P<0.05 (Tukey test).

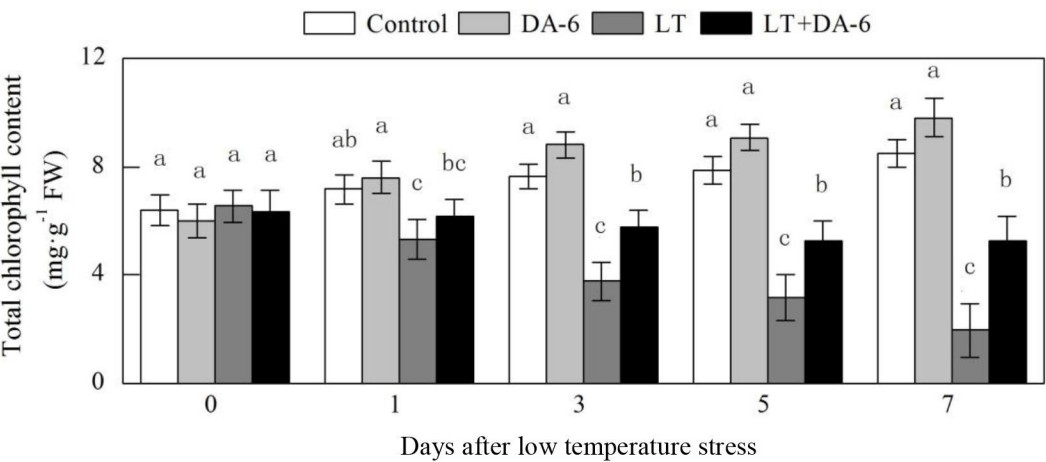

**Fig 4. Effects of LT and/or exogenous DA-6 on the total chlorophyll content.** The values represent the mean ± SE (n = 5), and values with the same letters in the columns are not significantly different at P<0.05 (Tukey test).

levels were downregulated by 66.19%, 76.14%, 76.75% and 72.99% in LT treatment; by 32.61%, 26.40%, 35.75% and 37.97% in LT +DA-6 treatment, respectively; NRT 1;2 relative expression levels were downregulated by 51.85%, 56.06%, 57.10% and 61.76% in LT treatment; by 27.33%, 19.85%, 30.65% and 35.25% in LT +DA-6 treatment, respectively; NRT 2;5 relative expression levels were downregulated by 53.38%, 48.58%, 61.40% and 69.18% in LT treatment; by 26.31%, 29.15%, 28.60% and 30.42% in LT +DA-6 treatment, respectively, compared with the levels in the control. Exogenous DA-6 significantly upregulated NRT 1;2 relative expression levels on the 5th day by 30.73%, compared with the levels in the control.

## Effects of LT and/or exogenous DA-6 on total chlorophyll content

Exogenous DA-6 partially suppressed the decrease in total chlorophyll content during LT (Fig 4). On the 1st, 3rd, 5th and 7th days, the total chlorophyll content decreased by 24.13%, 50.56%, 59.86% and 76.98% in the LT treatment and decreased by 12.42%, 24.69%, 32.70% and 37.70% in the LT+DA-6 treatment, respectively, compared with the levels in the control. Exogenous DA-6 increased the total chlorophyll content on the 3rd, 5th and 7th days by 4.91%, 9.92% and 9.66%, respectively, compared with the levels in the control.

## Effects of LT and/or exogenous DA-6 on gas exchange parameters

Exogenous DA-6 partially suppressed Pn, Gs and Tr during LT (Fig 5). On the 1st, 3rd, 5th and 7th days, Gs decreased by 44.27%, 55.97%, 64.85% and 73.55% in the LT treatment and decreased by 30.63%, 39.27%, 37.41% and 42.21% in the LT +DA-6 treatment, respectively; Tr decreased by 32.44%, 43.68%, 36.63% and 48.52% in the LT treatment and decreased by 13.92%, 21.34%, 13.52% and 25.16% in the LT+DA-6 treatment, respectively; and Pn decreased by 22.35%, 37.40%, 43.52% and 49.08% in the LT treatment and decreased by 18.78%, 23.48%, 26.03% and 32.06% in the LT+DA-6 treatment, respectively, compared with the levels in the control.

Over 7 days of LT, Ci decreased during the early period and then gradually increased, and minimum Ci levels were observed on the 3rd day. Ci decreased by 29.02% and 30.90% on the 1st and 3rd days, respectively, and increased by 12.49% on the 7th day in the LT treatment. Ci decreased by 24.82%, 23.28%, 21.32% and 14.80% on the 1st, 3rd, 5th and 7th days in the LT +DA-6 treatment, respectively, compared with the levels in the control. DA-6 had no significant effect on Ci under non-LT conditions.

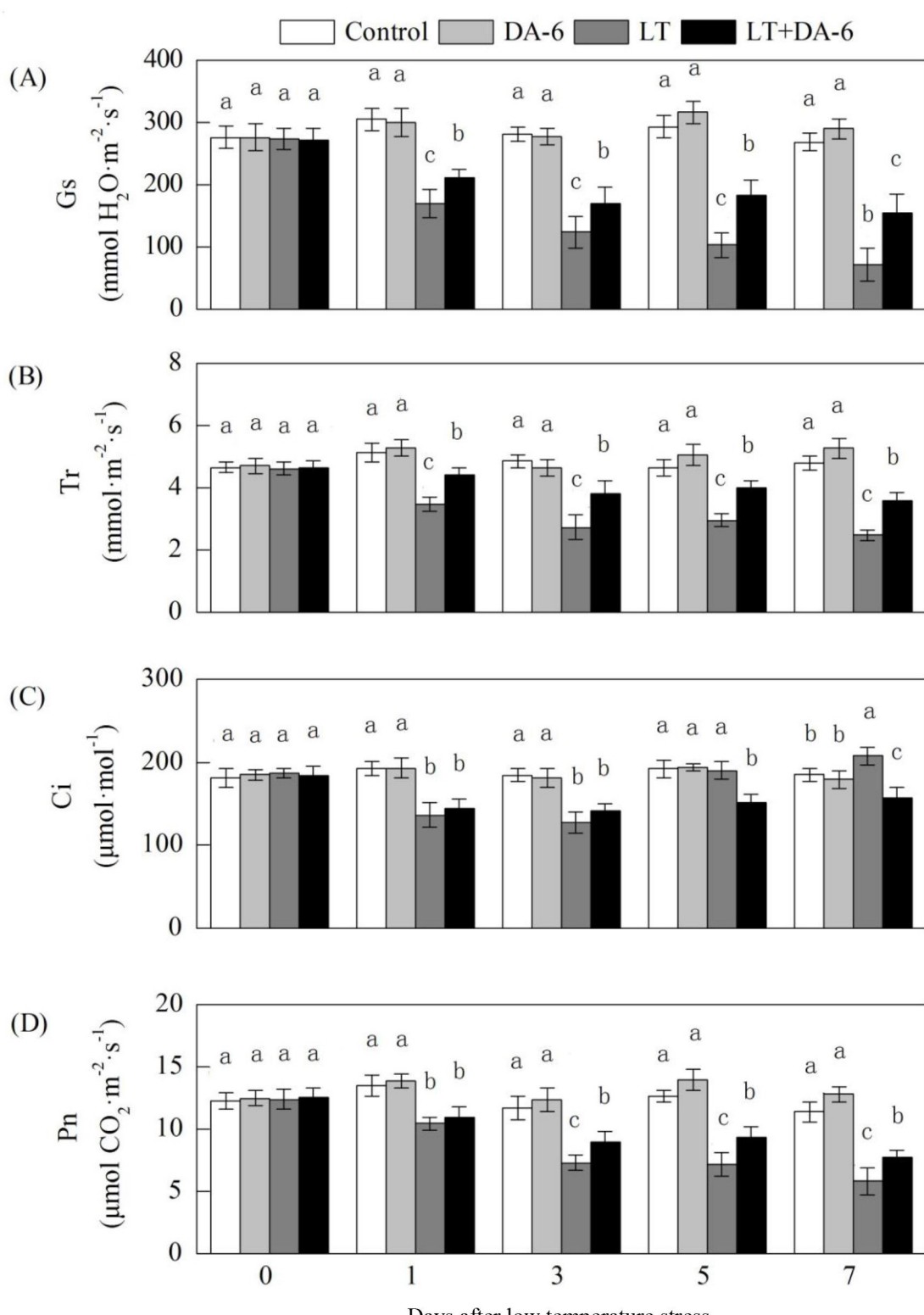

**Fig 5.** Effects of LT and/or exogenous DA-6 on Pn (A), Gs (B), Tr (C) and Ci (D). The values represent the mean ± SE (n = 5), and values with the same letters in the columns are not significantly different at P<0.05 (Tukey test).

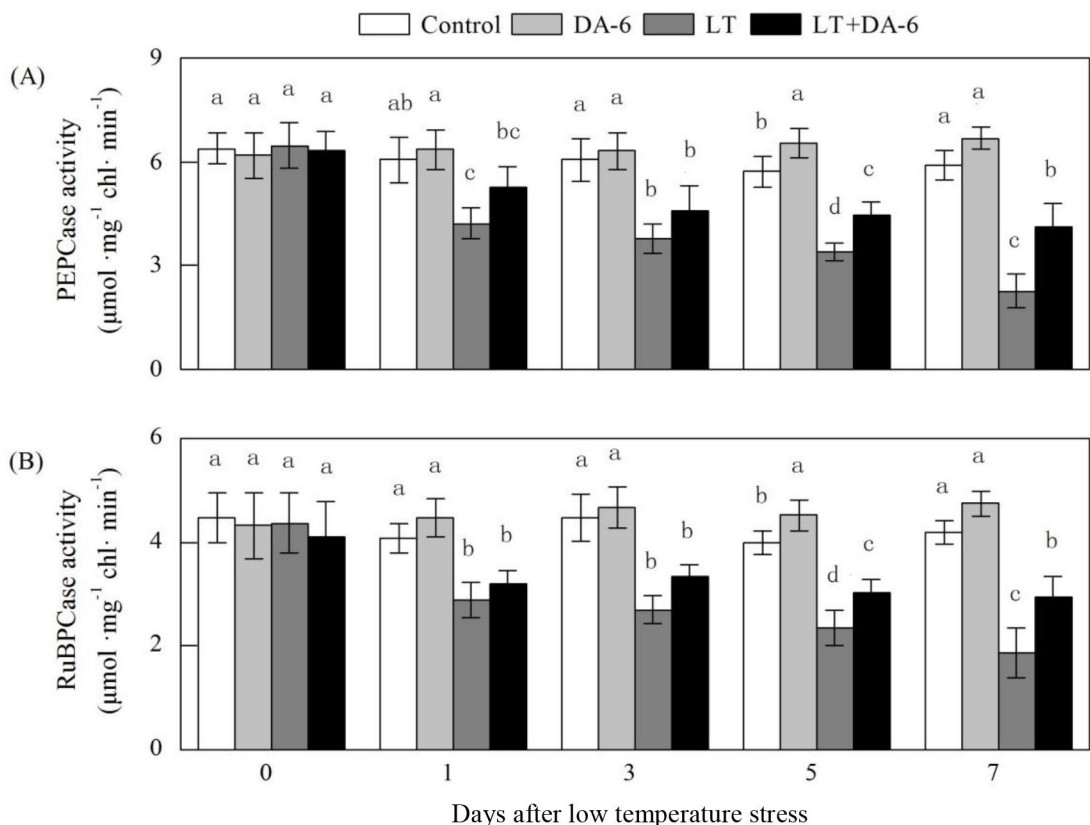

**Fig 6.** Effects of LT and/or exogenous DA-6 on the activities of PEPcase (A) and RuBPcase (B). The values represent the mean ± SE (n = 5), and values with the same letters in the columns are not significantly different at P<0.05 (Tukey test).

### Effects of LT and/or exogenous DA-6 on PEPcase and RuBPcase activities

The application of DA-6 mitigated the LT-induced reduction in PEPcase and RuBPcase activities in maize leaves over the experimental period (Fig 6). On the 1st, 3rd, 5th and 7th days, PEPcase activity decreased by 30.32%, 37.69%, 40.77% and 61.67% in the LT treatment, decreased by 13.21%, 23.87%, 21.82% and 30.21% in the LT+DA-6 treatment, and increased by 4.89%, 4.30%, 14.61% and 13.04% in the DA-6 treatment, respectively; RuBPcase activity decreased by 29.44%, 39.55%, 41.18% and 55.51% in the LT treatment, decreased by 21.34%, 25.50%, 23.99% and 29.80% in the LT+DA-6 treatment, and increased by 9.67%, 4.47%, 12.94% and 13.11% in the DA-6 treatment, respectively, compared with the levels in the control.

### Effects of LT and/or exogenous DA-6 on $O_2 \cdot^-$ generation rate and $H_2O_2$ content

Exogenous DA-6 partially suppressed increases in the $O_2 \cdot^-$ generation rate and $H_2O_2$ content during the LT treatments (Fig 7). On 1st, 3rd, 5th and 7th days, the $O_2 \cdot^-$ generation rate increased by 134.09%, 132.85%, 193.30% and 127.05% in the LT treatment and increased by 83.86%, 91.50%, 118.22% and 70.71% in the LT+DA-6 treatment, respectively; $H_2O_2$ content increased by 102.44%, 98.98%, 103.13% and 111.41% in the LT treatment and increased by 63.05%, 39.93%, 50.61%, 39.22% in the LT+DA-6 treatment, respectively, compared with the levels in the control.

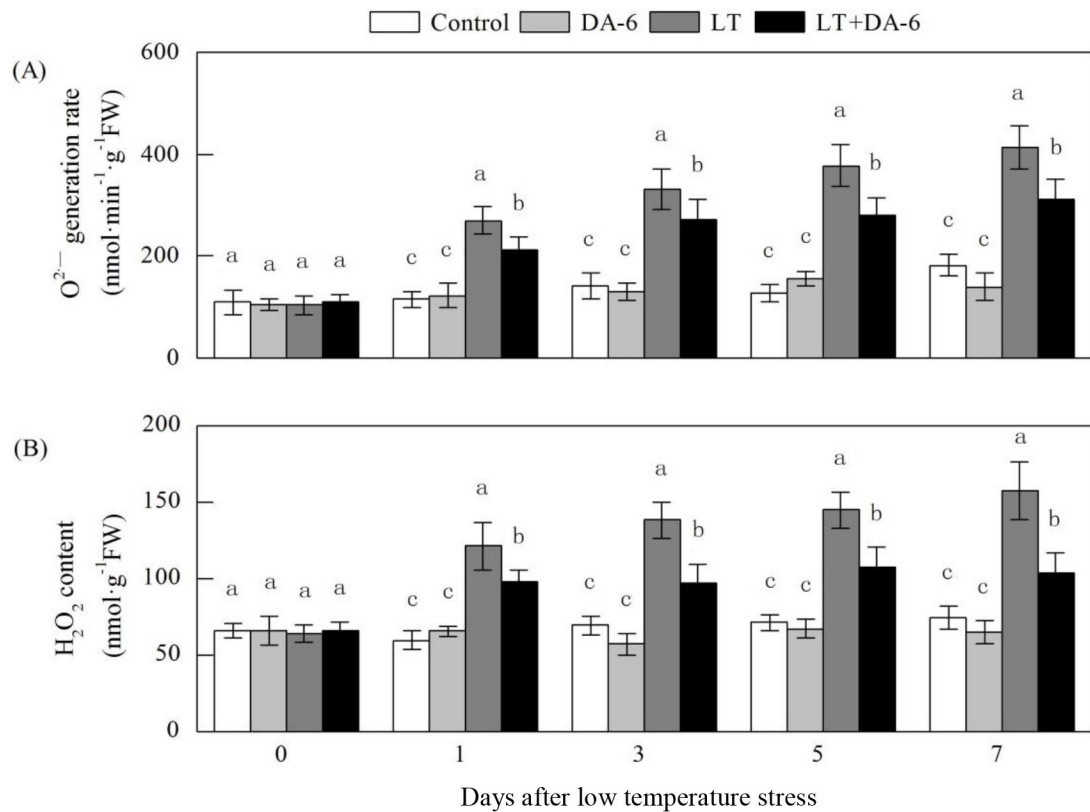

**Fig 7.** Effects of LT and/or exogenous DA-6 on the activities of SOD (A), POD (B) and CAT (C). The values represent the mean ± SE (n = 5), and values with the same letters in the columns are not significantly different at P<0.05 (Tukey test).

## Effects of LT and/or exogenous DA-6 on SOD, POD, CAT and APX activities

The activities of SOD and APX first increased and then declined slowly with the increasing duration of LT (Fig 8). Compared with the levels in the control, SOD activity increased by 139.40%, 89.74% and 56.83% on the 1st, 3rd and 5th days and decreased by 48.83% on the 7th day in the LT treatment; and increased by 133.42%, 108.85%, 75.18% and 79.50% on the 1st, 3rd, 5th and 7th days in the LT+DA-6 treatment, respectively; APX activity increased by 133.43% and 79.35% on the 1st and 3rd days, decreased by 16.97% and 44.75% on the 5th and 7th days in the LT treatment, and increased by 139.40%, 120.95%, 119.88% and 36.29% on the 1st, 3rd, 5th and 7th days in the LT+DA-6 treatment, respectively.

Over 7 days of LT, the activities of POD and CAT decreased gradually. On the 1st, 3rd, 5th and 7th days, compared with the control, POD activity decreased by 44.29%, 46.65%, 52.52% and 66.71% in the LT treatment and by 33.83%, 43.90%, 42.05% and 54.03% in the LT+DA-6 treatment, respectively; CAT activity decreased by 26.08%, 36.72%, 52.25% and 68.85% in the LT treatment and by 25.01%, 24.37%, 38.60% and 48.11% in the LT+DA-6 treatment, respectively. Exogenous DA-6 had no significant effect on the activities of POD and CAT.

## Effects of LT and/or exogenous DA-6 on $NO_3^-$, $NO_2^-$ and $NH_4^+$ contents and the $NO_3^-$ uptake rate

The decreases in $NO_3^-$ and $NO_2^-$ contents and the increases in $NH_4^+$ content were significantly suppressed by DA-6 under LT conditions (Fig 9). On the 1st, 3rd, 5th and 7th days, the

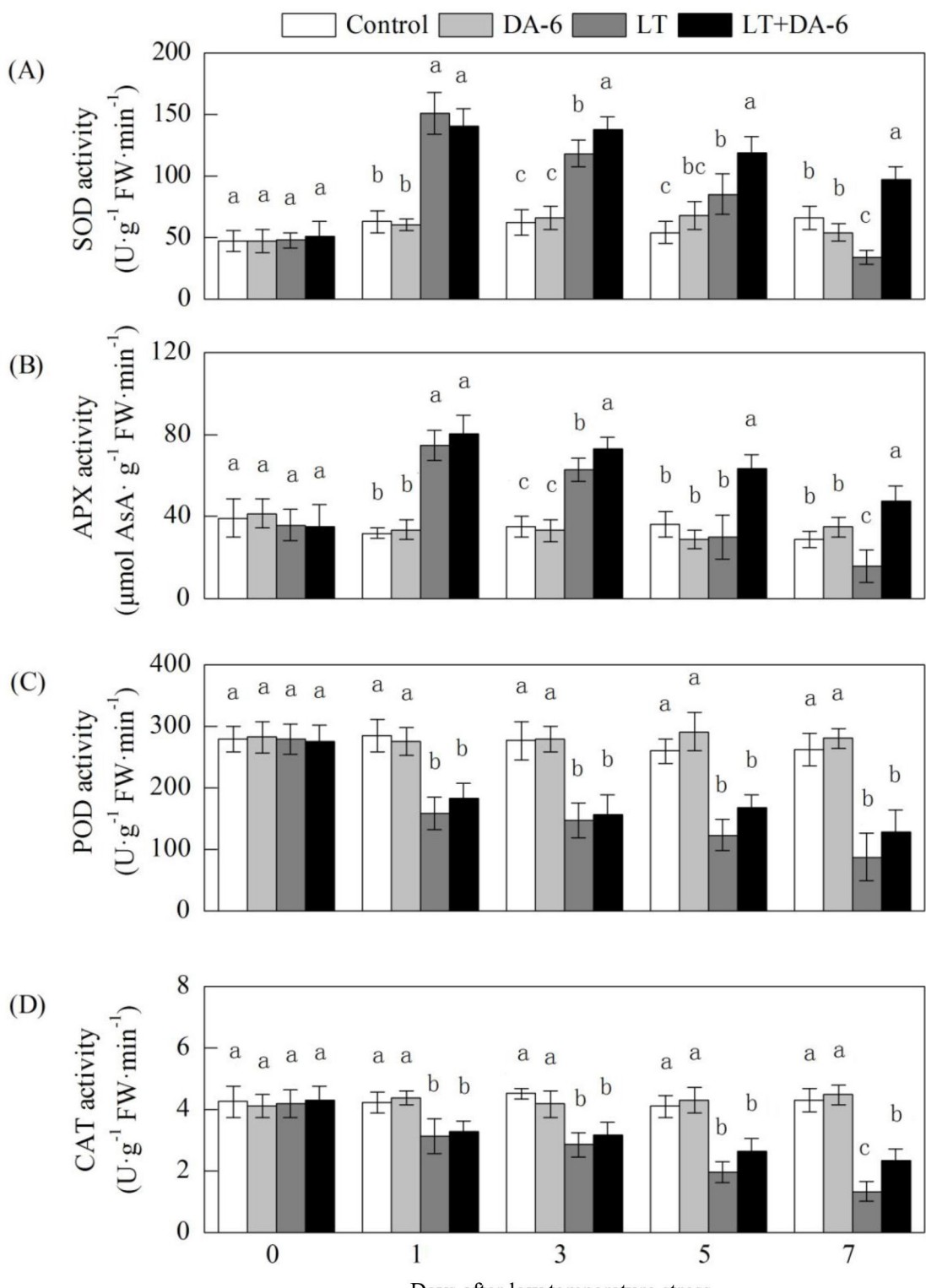

**Fig 8.** Effects of LT and/or exogenous DA-6 on the activities of SOD (A), POD (B), CAT (C) and APX (D). The values represent the mean ± SE (n = 5), and values with the same letters in the columns are not significantly different at P<0.05 (Tukey test).

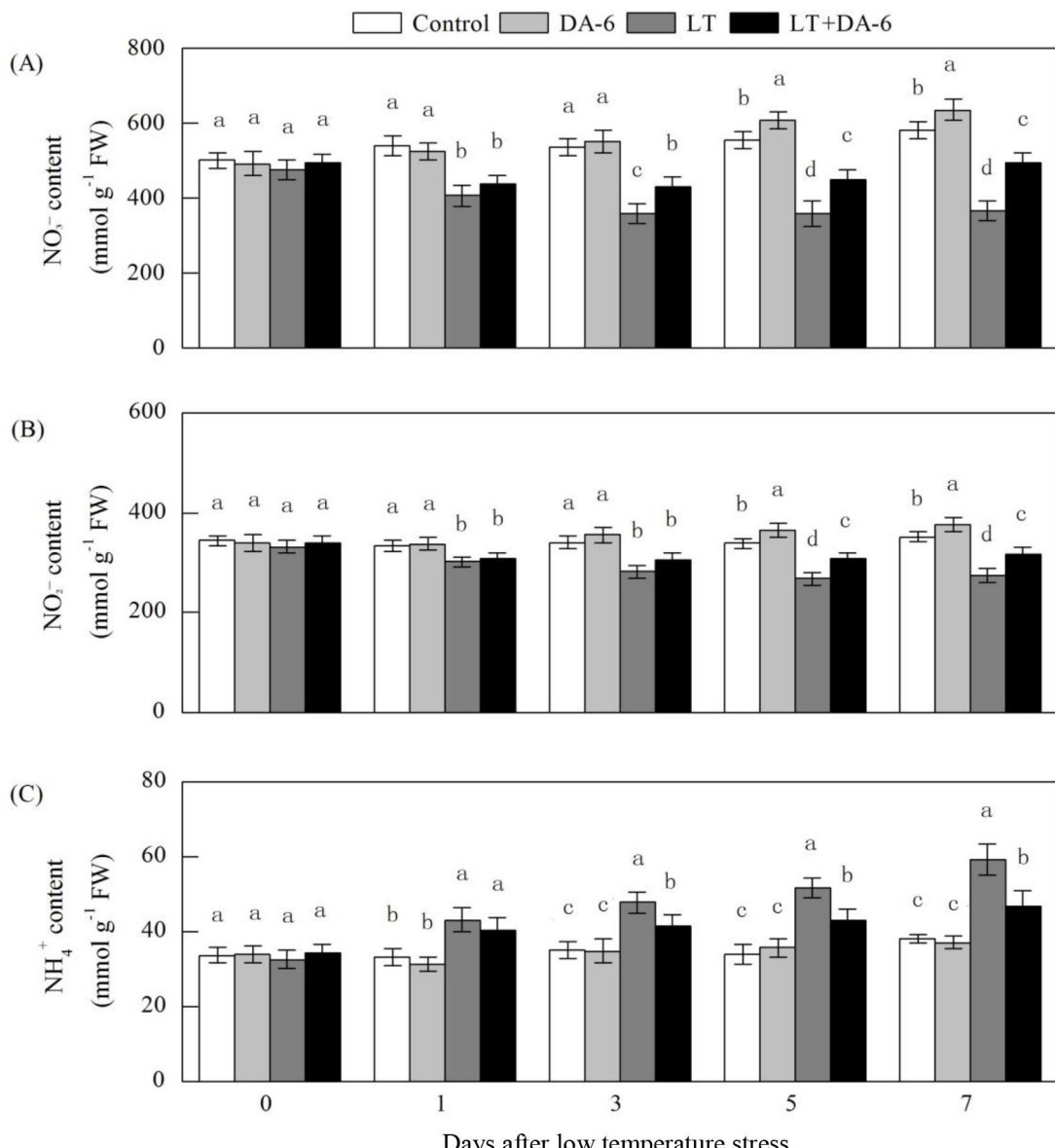

**Fig 9.** Effects of LT and/or exogenous DA-6 on the contents of $NO_3^-$ (A), $NO_2^-$ (B) and $NH_4^+$ (C). The values represent the mean $\pm$ SE (n = 5), and values with the same letters in the columns are not significantly different at P<0.05 (Tukey test).

$NO_3^-$ content decreased by 24.71%, 33.18%, 35.27% and 36.82% in the LT treatment and by 19.34%, 20.16%, 19.26% and 15.06% in the LT + DA-6 treatment, respectively; the $NO_2^-$ content decreased by 9.44%, 16.91%, 20.81% and 21.81% in the LT treatment and by 7.16%, 10.19%, 9.21% and 9.71% in the LT + DA-6 treatment, respectively. On the 5th and 7th days, the contents of $NO_3^-$ (increased by 9.78% and 9.45%, respectively) and $NO_2^-$ (increased by 7.67% and 7.50%, respectively) were significantly increased compared with the levels in the control. On the 1st, 3rd, 5th and 7th days, the $NH_4^+$ content increased by 30.15%, 36.32%, 52.51% and 55.02% in the LT treatment and by 22.37%, 17.81%, 26.50% and 22.43% in the LT + DA-6 treatment, respectively, compared with that of the control. No significant differences in the $NH_4^+$ contents were observed between the control and DA-6 treatments.

## Effects of LT and/or exogenous DA-6 on the activities of NR and NiR

Under LT conditions, the activities of foliar NR and NiR decreased initially and then remained stable (Fig 10). The activities of foliar NR and NiR were significantly increased by DA-6 under the same conditions. On the 1st, 3rd, 5th and 7th days, the NR activity decreased by 34.21%, 55.84%, 65.48% and 60.49% in the LT treatment and by 28.95%, 31.17%, 34.52% and 25.93% in the LT + DA-6 treatment, respectively; the NiR activity decreased by 45.04%, 73.22%, 83.71% and 78.83% in the LT treatment and by 38.17%, 40.88%, 44.15% and 46.45% in the LT + DA-6 treatment, respectively, compared with the levels in the control. NR activity increased by 14.29%, 19.05% and 17.28% and NiR activity increased by 18.77%, 24.35% and 18.36% on the 3rd, 5th and 7th days, respectively, compared with the levels in the control.

## Effects of LT and/or exogenous DA-6 on GS, GOGAT and GDH activities

Under LT conditions, the activities of GS and GOGAT in the leaves decreased during the early period and then remained stable (Fig 11). The activities of GS and GOGAT were significantly increased by DA-6 under the same conditions. On the 1st, 3rd, 5th and 7th days, the GS activity decreased by 20.01%, 33.97%, 41.16% and 36.04% under LT conditions and by 16.44%, 18.96%, 21.72% and 19.61% in the LT + DA-6 treatment; the GOGAT activity decreased by 30.54%, 49.90%, 59.06% and 55.97% in the LT treatment and by 25.87%, 27.86%, 31.15% and 33.52% in the LT + DA-6 treatment, respectively. GS activity increased by 8.68%, 7.26% and 9.71% and GOGAT activity significantly increased by 12.79%, 12.08% and 11.41% on the 3rd, 5th and 7th days in the DA-6 treatment, respectively, compared with the levels in the control.

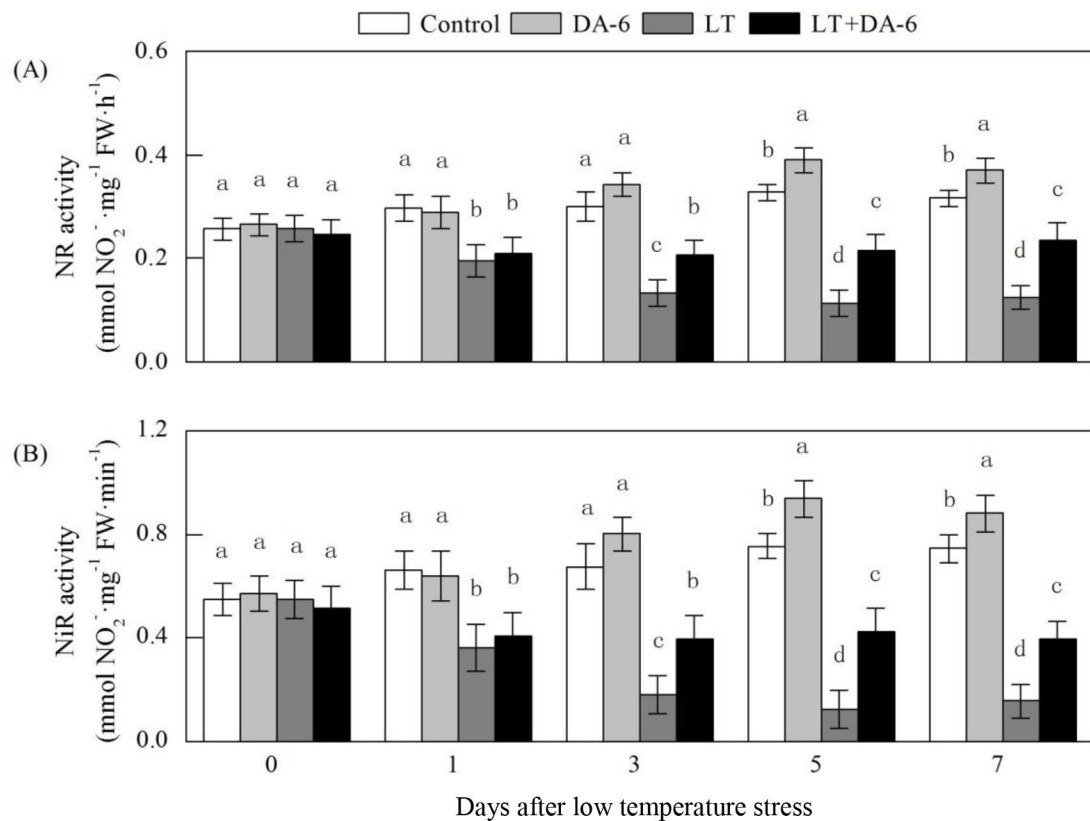

**Fig 10.** Effects of LT and/or exogenous DA-6 on the activities of NR (A) and NiR (B). The values represent the mean ± SE (n = 5). Values with the same letters in the columns are not significantly different at P<0.05 (Tukey test).

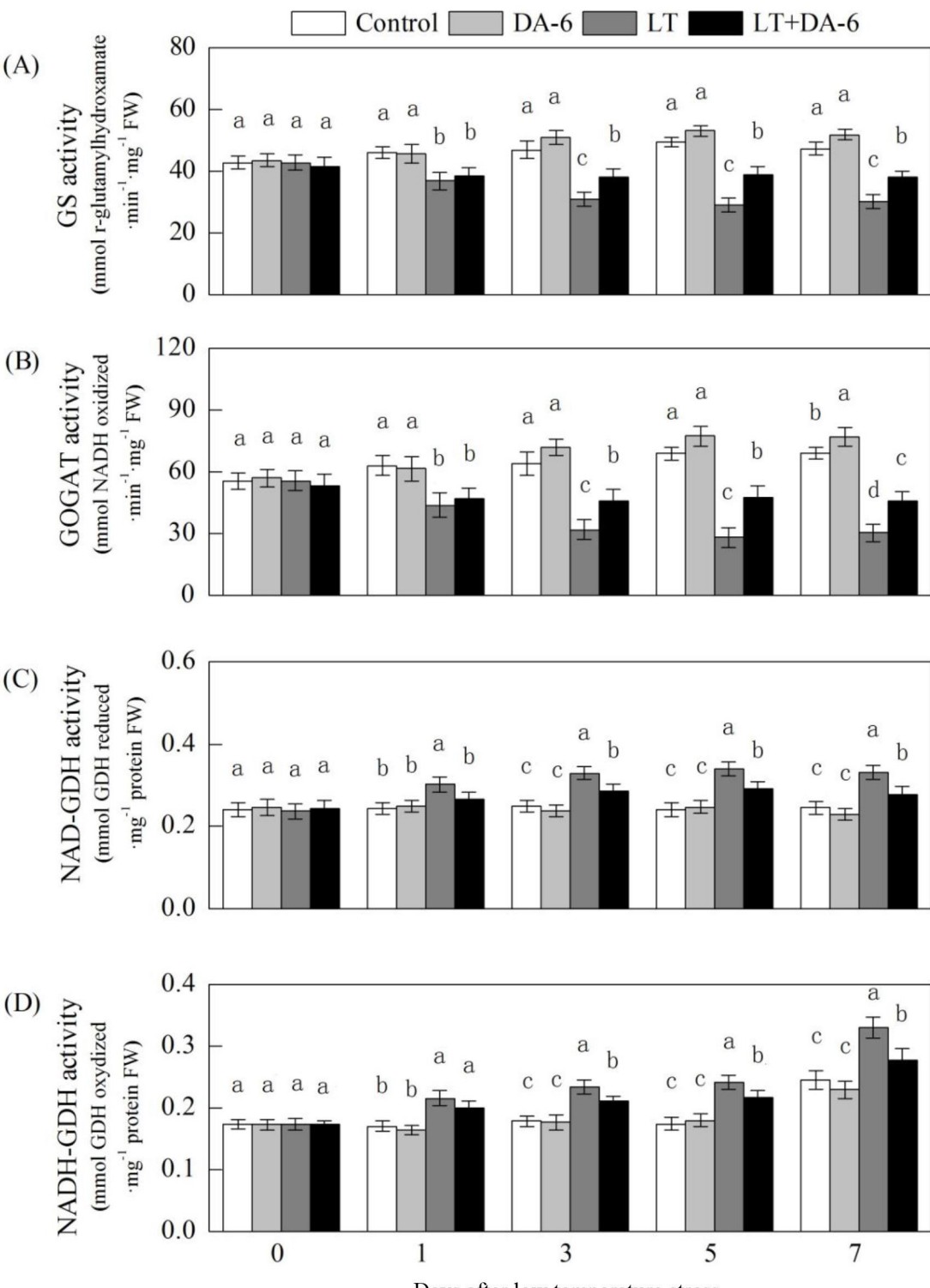

**Fig 11.** Effects of LT and/or exogenous DA-6 on the activities of GS (A), GOGAT (B), NAD-GDH (C) and NADH-GDH (D). The values represent the mean ± SE (n = 5), and values with the same letters in the columns are not significantly different at $P < 0.05$ (Tukey test).

In contrast, LT enhanced the activities of NAD-GDH and NADH-GDH: NAD-GDH activity increased by 24.22%, 31.97%, 41.45% and 34.76% and NADH-GDH activity increased by 26.37%, 31.16%, 38.99% and 34.76%, respectively, compared with that in the control on the 1st, 3rd, 5th and 7th days. However, NAD-GDH activity was increased by 9.49%, 15.20%, 21.27% and 13.43%, and NADH-GDH activity was increased by 17.21%, 18.48%, 24.05% and 13.43% in the LT+ DA-6 treatment compared with that in the control on the 1st, 3rd, 5th and 7th days.

### Effects of LT and/or exogenous DA-6 on AlaAT and AspAT activities

The AlaAT and AspAT activities decreased after LT treatment (Fig 12). Upon DA-6 application, the AlaAT and AspAT activities were all significantly elevated, especially in the stressed plants. On the 1st, 3rd, 5th and 7th days, AlaAT activity decreased by 13.50%, 22.35%, 32.62% and 32.93% in the LT treatment, decreased by 4.37%, 8.27%, 14.62% and 16.64% in the LT +DA-6 treatment, and increased by 3.22%, 8.10%, 10.95% and 9.79% in the DA-6 treatment, respectively; AspAT activity decreased by 30.12%, 44.73%, 48.24% and 56.57% in the LT treatment, decreased by 22.58%, 28.38%, 24.50% and 34.21% in the LT+DA-6 treatment, and increased by 4.24%, 10.39%, 14.77% and 10.53% in response to the DA-6 treatment, respectively, compared with the levels in the control.

### Effects of LT and/or exogenous DA-6 on free amino acid and soluble protein contents and proteinase activity

No difference in the contents of free amino acids and soluble protein or in proteinase activity was noted between the DA-6 application and the non-DA-6 application under LT conditions

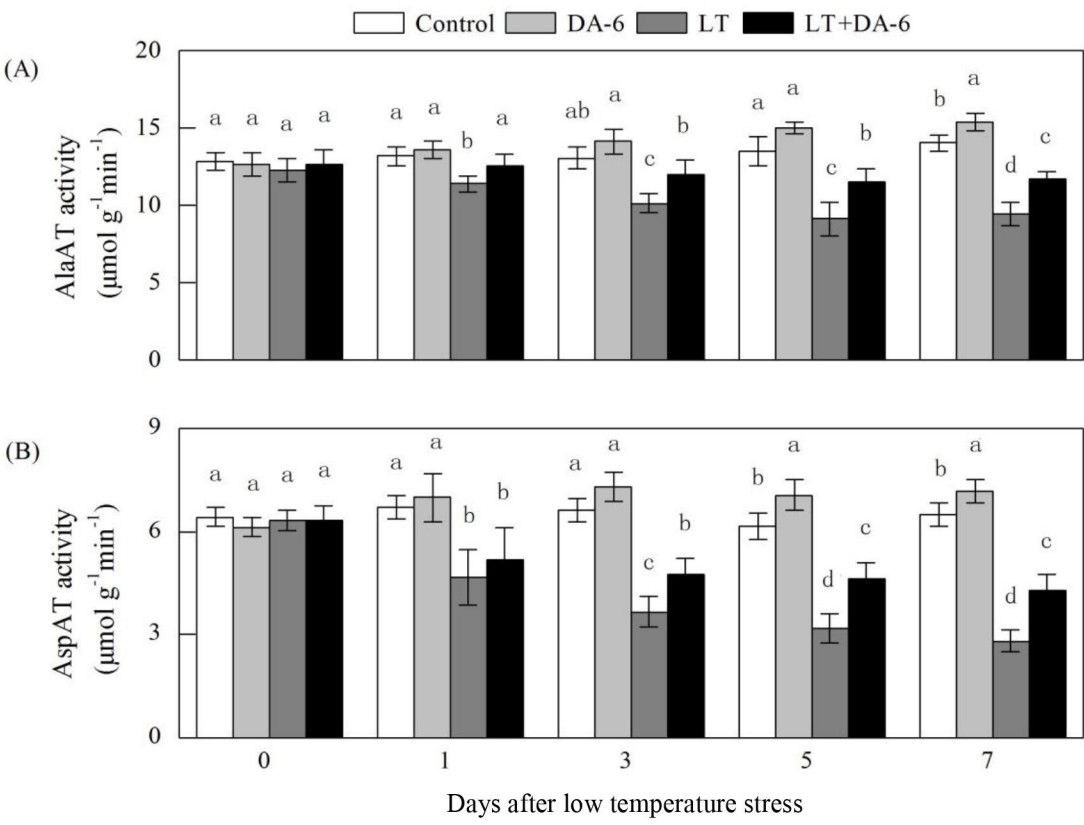

**Fig 12.** Effects of LT and/or exogenous DA-6 on the activities of AlaAT (A) and AspAT (B). The values represent the mean ± SE (n = 5), and values with the same letters in the columns are not significantly different at P<0.05 (Tukey test).

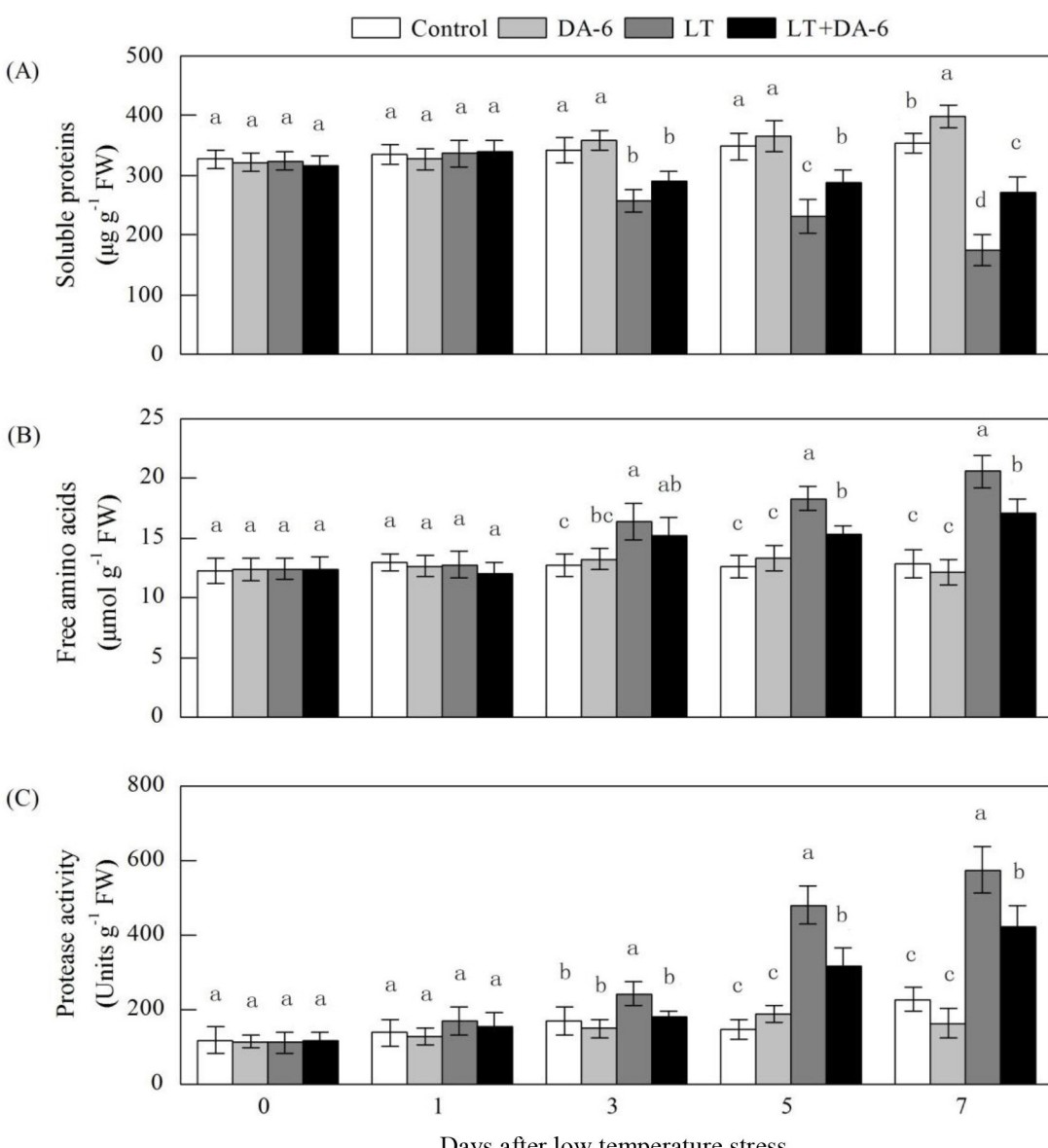

**Fig 13.** Effects of LT and/or exogenous DA-6 on the free amino acid content (A), soluble protein content (B), and proteinase activity (C). The values represent the mean ± SE (n = 5), and values with the same letters in the columns are not significantly different at P<0.05 (Tukey test).

on the 1st day (Fig 13). DA-6 application suppressed the increase in the free amino acid content and proteinase activity and the decrease in the soluble protein content that were induced by LT. On the 3rd, 5th and 7th days, the free amino acid content increased by 28.25%, 44.95% and 60.06% in the LT treatment and increased by 8.27%, 14.62% and 16.64% in the LT+DA-6 treatment, respectively; the soluble protein content decreased by 24.65%, 33.72% and 50.78% in the LT treatment, decreased by 15.32%, 17.73% and 23.36% in the LT+DA-6 treatment, and increased by 4.94%, 4.97% and 12.14% in the DA-6 treatment, respectively; the proteinase activity increased by 43.88%, 225.13% and 152.12% in the LT treatment and decreased by 6.91%, 113.61% and 84.67% in the LT+DA-6 treatment, respectively, compared with the levels in the control.

## Discussion

The growth inhibition of maize seedlings under LT conditions in the present study has been observed previously and may be attributable to a reduction in cell enlargement and cell division induced by the stunting of physiological activities [36,37]. Similar to prior findings, exogenous DA-6 promoted the growth of maize seedlings under non-stress conditions in this study [10]. Moreover, the growth inhibition of maize seedlings induced by LT was partially counteracted by exogenous DA-6 application, as demonstrated by the significantly increased growth parameters compared with those under LT conditions (Table 1). These results suggested that exogenous DA-6 could enhance the LT tolerance of maize seedlings.

Although various forms of N, such as $NO_3^-$, $NH_4^+$ and amino acids, are available for metabolic processes, $NO_3^-$ is the predominant N form used by crops. As in previous reports, LT significantly diminished the foliar $NO_3^-$ content in this study (Fig 9A) [38]. This may be attributed to the suppression of $NO_3^-$ absorption by the roots and the disturbance of $NO_3^-$ transport in xylem under LT conditions.

Plant roots are a vital organ system for water and nutrient acquisition and play critical roles in plant adaptations to stress [39]. In the root system, NRT protein family members are responsible for $NO_3^-$ transport. Wang et al. showed that water limitation enhanced the expression of NRT1;2 and NRT2;5 but had no significant effect on NRT1;1 expression [40]. In the present study, the relative expression levels of NRT1;1, NRT1;2 and NRT2;5 were downregulated by LT (Fig 3). These results suggested that the regulation of the relative expression levels of NRT1;1, NRT1;2 and NRT2;5 may be related to the types and intensities of stresses. The inhibition of root growth and the downregulated NRT1;1, NRT1;2 and NRT2;5 relative expression levels may be the main cause of the suppression of $NO_3^-$ uptake in roots. In the LT +DA-6 treatment, the decrease in foliar $NO_3^-$ content was partly alleviated by exogenous DA-6. This can primarily be attributed to the improved uptake of $NO_3^-$ due to the upregulated expression levels of NRT1;1, NRT1;2, and NRT2;5 and the larger root system, as measured by the increased length, surface area, and volume of roots.

Once taken up into the roots, $NO_3^-$ undergoes long-distance transport to the leaves. This process depends on transpiration intensity and root pressure. In this study, exogenous DA-6 reduced the continued decline of Gs during LT and maintained the transport of water and nutrients in xylem sap from the roots to the leaves of the maize seedlings (Fig 5B). This may be due to the balance between water loss by transpiration and water uptake from the extensive root system induced by the exogenous DA-6 [41]. In addition, exogenous DA-6 increased Lp, which may be due to the enhanced physiological activity of the whole root system. The higher transpiration occurred in conjunction with greater Lp, leading to the significantly improved transport of $NO_3^-$ from roots to shoots during LT.

$NO_3^-$ is the only storage form of N and is converted to $NH_4^+$ prior to its incorporation into amino acids [42]. $NO_3^-$ is first reduced to $NO_2^-$ by NR; this process is highly sensitive to environmental stress conditions and is considered the rate-limiting step in $NO_3^-$ assimilation [43]. NR activity is primarily regulated by the $NO_3^-$ concentration and is very sensitive to $H_2O_2$ [44]. In this study, NR activity continuously declined under LT conditions (Fig 10A). This may be attributed to the decrease in foliar $NO_3^-$ concentration and the excessive accumulation of $H_2O_2$ due to the imbalanced generation and scavenging of ROS under LT conditions (Figs 7B and 9A). Interestingly, the negative effect of LT on NR was ameliorated to some extent by exogenous DA-6. One possible reason is the increased foliar $NO_3^-$ content induced by exogenous DA-6. Plants can defend against oxidative stress through the combined action of enzymatic and nonenzymatic antioxidants to eliminate ROS. $O_2 \cdot^-$ is converted into $O_2$ and $H_2O_2$ by SOD as the first step in ROS scavenging. Then, $H_2O_2$ is further detoxified via conversion

into $H_2O$ by antioxidant enzymes, such as POD and CAT, as well as the ascorbate-glutathione (AsA-GSH) cycle. In this study, the decrease in foliar $O_2^{.-}$ and $H_2O_2$ accumulation in the DA-6+LT treatment may be attributed to the enhanced antioxidant protection of the increased activities of SOD, POD, CAT and APX in the leaves of the maize seedlings. The decreased foliar $H_2O_2$ accumulation may partly contribute to the increased NR activity in plants under LT conditions (Fig 8). The reduction in foliar $NO_2^-$ content was significantly reversed by exogenous DA-6 under LT conditions, which may be a result of the increases in both the foliar $NO_3^-$ content and the NR activity. Subsequently, $NO_2^-$ is reduced to $NH_4^+$ by NiR. As in NR, a significant reduction in NiR activity was also noted under LT conditions. This inhibition was associated with reductions in both the $NO_3^-$ and $NO_2^-$ contents. Exogenous DA-6 upregulated NiR activity, which was downregulated by LT, and promoted the conversion of $NO_2^-$ to $NH_4^+$. These results suggest that exogenous DA-6 could effectively regulate the activities of NR and NiR and maintain the $NO_3^-$ assimilation process under LT conditions.

$NH_4^+$, formed by the disruption of $NO_3^-$ assimilation and the hydrolysis of N-containing metabolites, is harmful to cells and must be quickly assimilated. For higher plants, the GS/GOGAT cycle is the main $NH_4^+$ assimilation pathway [45]. In this cycle, $NH_4^+$ is converted to glutamine by GS and then to glutamate by GOGAT, which is integrated directly into the structures of amino acids. Although LT decreased NR and NiR activities, the $NH_4^+$ content significantly increased compared with that in the plants under LT conditions. The activities of GS and GOGAT decreased in a range of plants in response to a variety of environmental stresses. Since GS is the primary enzyme responsible for $NH_4^+$ assimilation in plants, the reduced GS activity induced by LT might result in a partial increase in the foliar $NH_4^+$ content (Fig 11). The decline in GOGAT activity found in the LT-treated plants could have a detrimental impact on the conversion of glutamine to glutamate in leaves. The reductions in both GOGAT and GS activities observed in LT-stressed plants may be partly attributed to oxidative modifications of enzyme proteins [46]. However, the foliar $NH_4^+$ content was reduced in the LT+DA-6 treatment, which may be attributed to the increased activities of GS and GOGAT, which promoted the integration of $NH_4^+$ into the structure of organic compounds.

In green tissues, GOGAT and GS obtain reducing power directly from photosynthesis [47]. Exogenous DA-6 mitigated the reductions in the total chlorophyll content under LT conditions and maintained the quantum harvesting ability of the leaves, thereby contributing to the maintenance of a more efficient process in the light reactions of photosynthesis and supplying GOGAT and GS with sufficient reducing powers in the form of NADPH, ATP, or $Fd_{red}$. The results suggested that the effective GOGAT/GS cycle in the maize seedling leaves under LT conditions was potentially attributable to the ameliorated photosynthesis. In addition, the restraint of $O_2^{.-}$ and $H_2O_2$ accumulation in the leaves of maize seedlings may be another cause for the stable GS and GOGAT activity in the LT+DA-6 treatment.

As in previous studies on maize seedlings, exogenous DA-6 also enhanced Rubisco and PEPCase activities under non-stress conditions in this study [10]. Moreover, seedlings treated with DA-6 maintained stable PEPCase and RuBPCase activities during LT. This may be due to the GOGAT/GS cycle effectively removing the toxic $NH_4^+$ derived from photorespiration to protect the photosynthetic enzymes, promoting the fixation of atmospheric $CO_2$ into oxaloacetate through the carboxylation of phosphoenolpyruvate and the released of $CO_2$ re-fixed during the Calvin cycle, and causing an increase in the $CO_2$ assimilation capacity which was inhibited by LT, as observed in this study and a previous study [48]. The carbohydrates generated through photosynthesis are major building blocks and energy sources for biomass production and maintenance. The growth promotion of maize seedlings treated with DA-6 may be partly attributed to the stable photosynthetic capacity under LT conditions.

When the GS/GOGAT cycle pathway is inhibited under stress conditions, the $NH_4^+$ content in the plant cells increases considerably. $NH_4^+$ can serve as a substrate for glutamate formation via the reversible amination of 2-oxoglutarate through the catalytic effect of the GDH enzyme, although GDH has a lower affinity for $NH_4^+$ [49]. The increased activity of GDH during the early period of LT promoted the conversion of $NH_4^+$ to glutamate and alleviated $NH_4^+$ toxicity. However, GDH activity was subsequently decreased, accompanied by the $NH_4^+$ content increasing considerably (Fig 4E and 4F). DA-6 application reduced NADH-GDH activity and $NH_4^+$ content, which may be associated with enhanced GS and GOGAT activities. These results suggested that exogenous DA-6 could effectively regulate the activities of GS, GOGAT and GDH and maintain the conversion of $NH_4^+$ to glutamate under LT conditions.

Glutamate produced by the GS/GOGAT cycle and the GDH pathway is the primary amino acid responsible for the synthesis of other amino acids [50]. Transamination reactions, which transfer amino groups from glutamate to other amino acids, serve as a link between carbohydrate and amino acid metabolism and are essential for plant growth. In this study, the stress-induced decreases in foliar AlaAT and AspAT activities may be attributable to the weakened GS/GOGAT pathway (Fig 12) [51]. Moreover, exogenous DA-6 inhibited the reduction in the AlaAT and AspAT activities induced by LT to some extent. This finding may be associated with increased GS/GOGAT activities, which generate more glutamate to serve as a substrate for transamination reactions in maize seedlings treated with DA-6 under LT. Therefore, exogenous DA-6 could effectively regulate AlaAT and AspAT activities and promote the formation of alanine from pyruvate and glutamate, the synthesis of aspartate from glutamate and oxaloacetate, and subsequently the synthesis of other amino acids.

Most soluble proteins are enzymes that participate in various metabolic pathways in plants; therefore, the soluble protein content is considered one of the most important indices reflecting the overall metabolic level in plants [52]. Protein synthesis in plants is very sensitive to abiotic stresses and is positively correlated with stress tolerance [53]. In the present study, the amount of foliar soluble protein significantly decreased after LT exposure on the 1$^{st}$ day compared with that in the control (Fig 13). Possible explanations include the following: the protease activity was enhanced [54], the stability of proteins was altered [55], and/or protein degradation occurred due to the toxic effects of ROS induced by stress [56]. The maize seedlings treated with DA-6 maintained higher levels of soluble protein than the non-DA-6-treated seedlings in response to LT. The results suggested that exogenous DA-6 might inhibit protein degradation and/or accelerate the synthesis process of some original proteins; subsequently, the treated plants may maintain a certain turgor and further ensure that a series of physiological and biochemical processes occur normally under LT conditions. Free amino acids are the building blocks of proteins. The increased foliar free amino acid contents in plants exposed to LT may be attributed to increases in proteolysis or a decrease in protein synthesis (Fig 6B). DA-6 may promote the biosynthesis and accumulation of amino acids, which in turn may regulate osmotic adjustment, protect cellular macromolecules, store nitrogen, and maintain the cellular pH [57].

## Conclusion

Low temperature inhibited the growth of maize seedlings, disturbed the processes of nitrogen metabolism and photosynthesis, and induced oxidative stress. Under LT conditions, exogenous DA-6 enhanced $NO_3^-$ uptake by promoting root growth and stable relative expression levels of NRT1;1, NRT1;2 and NRT2;5; maintained the transport of $NO_3^-$ from roots to shoots by increasing Gs and Lp; and promoted the assimilation of $NO_3^-$ and the conversion of $NH_4^+$ to glutamate by effectively regulating the activities of NR, NiR, GS, GOGAT and GDH, which

may be associated with improved photosynthesis and antioxidant metabolism. In addition, exogenous DA-6 maintained transamination through stable AlaAT and AspAT activity and increased the protein content and decreased the free amino acid content under LT conditions to ensure normal growth. These results indicate that the negative effects of LT on maize seedling growth can be eased to some extent by exogenous DA-6.

## Supporting information

**S1 Table. Gene-specific primers used in the real-time polymerase chain reaction (PCR) analysis.**
(DOCX)

## Acknowledgments

We are grateful for the valuable comments by the editor and reviewers that improved this manuscript.

## Author Contributions

**Data curation:** Jianguo Zhang, Zhenhua Wang.

**Formal analysis:** Jianguo Zhang, Shujun Li.

**Funding acquisition:** Jianguo Zhang, Zhenhua Wang.

**Investigation:** Zhenhua Wang, Tao Yu.

**Methodology:** Jianguo Zhang, Shujun Li.

**Resources:** Jianguo Zhang.

**Software:** Shujun Li.

**Supervision:** Tao Yu.

**Validation:** Tao Yu.

**Visualization:** Shujun Li, Zhenhua Wang.

**Writing – original draft:** Jianguo Zhang.

**Writing – review & editing:** Quan Cai, Jingsheng Cao, Tenglong Xie.

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
