## [Decision Letter · Decision Letter 0]

10 Feb 2020

PONE-D-19-36030

Exogenous Diethyl Aminoethyl Hexanoate Ameliorates Low Temperature Stress by
Improving Nitrogen Metabolism in Maize Seedlings

PLOS ONE

Dear Dr. Wang,

Thank you for submitting your manuscript to PLOS ONE. After careful consideration, we
feel that it has merit but does not fully meet PLOS ONE’s publication criteria as it
currently stands. Therefore, we invite you to submit a revised version of the
manuscript that addresses the points raised during the review process.

ACADEMIC
EDITOR: After careful evaluation by myself and comments of the
worthy reviewers, I recommend a major revision in the current draft of ms,
particularly regarding data presentation and description.

We would appreciate receiving your revised manuscript by Mar 26 2020 11:59PM. When
you are ready to submit your revision, log on to https://www.editorialmanager.com/pone/ and select the 'Submissions
Needing Revision' folder to locate your manuscript file.

If you would like to make changes to your financial disclosure, please include your
updated statement in your cover letter.

To enhance the reproducibility of your results, we recommend that if applicable you
deposit your laboratory protocols in protocols.io, where a protocol can be assigned
its own identifier (DOI) such that it can be cited independently in the future. For
instructions see: http://journals.plos.org/plosone/s/submission-guidelines#loc-laboratory-protocols

We look forward to receiving your revised manuscript.

Kind regards,

Saddam Hussain

Academic Editor

PLOS ONE

Additional Editor Comments (if provided):

After careful evaluation by myself and comments of the worthy reviewers, I recommend
a major revision in the current draft of ms, particularly regarding data
presentation and description.

Journal Requirements:

Reviewers' comments:

Reviewer's Responses to Questions

**Comments to the Author**

1. Is the manuscript technically sound, and do the data support the conclusions?

Reviewer #1: Yes

Reviewer #2: Yes

2. Has the statistical analysis been performed
appropriately and rigorously? 

Reviewer #1: I Don't Know

Reviewer #2: Yes

3. Have the authors made all data underlying the
findings in their manuscript fully available?

Reviewer #1: Yes

Reviewer #2: Yes

4. Is the manuscript presented in an intelligible
fashion and written in standard English?

Reviewer #1: Yes

Reviewer #2: Yes

5. Review Comments to the Author

Reviewer #1: General comment:

Spring maize was vulnerable to low temperatures during sowing time in Northeast
China. The effects of 6-DA on N metabolism and photosynthesis in maize at low
temperature were studied, and the test indices were abundant (from root to leaf).
The application of 6-DA has important practical significance to address LT on spring
maize. Too many Figures and tables was a problem in this article, some test indices
seem to be able to be removed. Hence, we recommended this paper to publish in PLOS
One after minor revision.

Specific comment:

(1) Abstract:

Line 26-27, ‘DA-6 regulates many aspects… are unknown.’ This sentence is not good,
spring maize sowing time meet to the low temperature in northeast China, and the LT
suppressed the N metabolism and Pn, further reduced the dry matter accumulation.
DA-6 had the function of improving N metabolism, hence, we studied the effect of
DA-6 on maize in low temperature condition.

(2) Introduction:

Line 49-50, the frequency of low temperature occurrence during the sowing period of
spring maize was not detailed, and lacking meteorological data.

Line 59, the introduction of 6-DA is not detailed. As a growth regulator, what is its
corresponding plant hormone and what are its physiological functions and action
mechanisms.

(3) Materials and methods:

Line 122-127, the determination of APX is missing.

(4) Results:

Table 3 and Fig.1 can be merged, and the data of Table 3 should be added into
Fig.1.

Line 276, APX is missing.

Line 327, NAD-GDH (C), NADH-GDH (D)

(5) Figures caption:

Fig 2-Fig 12, name of abscissa ‘Days after stress’ should be ‘Days after low
temperature stress’

What’s the meaning of ‘b-d’ in the columns in Fig.2? Other Figures have the same
problem.

The letters in the columns of Fig.7 (A) is wrong, need to revise.

Reviewer #2: Please find the point-wise comments and suggestions in the attached file
and answer/address each one appropriately. The English language needs improvement
throughout the manuscript for clarity of reading.

6. PLOS authors have the option to publish the peer
review history of their article (what does this mean?). If published, this will
include your full peer review and any attached files.

If you choose “no”, your identity will remain anonymous but your review may still be
made public.

**Do you want your identity to be public for this peer review?** For
information about this choice, including consent withdrawal, please see our
Privacy Policy.

Reviewer #1: No

Reviewer #2: No

comment.docx

---

## [Author Response · Author response to Decision Letter 0]

24 Mar 2020

Dear Reviewers

I have completed the revision of our manuscript titled “Exogenous Diethyl Aminoethyl
Hexanoate Ameliorates Low Temperature Stress by Improving Nitrogen Metabolism in
Maize Seedlings” (PONE-D-19-36030), following two reviewers' recommendations. I have
incorporated the suggestions made by the two reviewers. I have benefited greatly
from the review process. Major changes are summarized below:

Response to Reviewer 1:

Comment: Too many Figures and tables was a problem in this article, some test indices
seem to be able to be removed. 

Reply: Results section, we have moved the Table 1 to the Supplementary 1.

1. Abstract:

Comment: Line 26-29, ‘DA-6 regulates many aspects… are unknown.’ This sentence is not
good, spring maize sowing time meet to the low temperature in northeast China, and
the LT suppressed the N metabolism and Pn, further reduced the dry matter
accumulation. DA-6 had the function of improving N metabolism, hence, we studied the
effect of DA-6 on maize in low temperature condition.

Reply: Line 26-29, page 2 in the revised manuscript, we have realized the sentence
“DA-6 regulates many aspects… are unknown.” is not good, and instead it by “Spring
maize sowing occurs during a period of low temperature (LT) in Northeast China, and
the LT suppresses nitrogen (N) metabolism and photosynthesis, further reducing dry
matter accumulation. Diethyl aminoethyl hexanoate (DA-6) improves N metabolism;
hence, we studied the effects of DA-6 on maize seedlings under LT conditions.”

2. Introduction:

Comment: Line 49-50, the frequency of low temperature occurrence during the sowing
period of spring maize was not detailed, and lacking meteorological data.

Reply: Lines 50-53, page 3 in the revised manuscript, we have detailed the relevant
information, and supplemented the meteorological data.

Comment: Line 59, the introduction of 6-DA is not detailed. As a growth regulator,
what is its corresponding plant hormone and what are its physiological functions and
action mechanisms.

Reply: Lines 62-71, pages 3 and 4 in the revised manuscript, we have supplemented the
corresponding plant hormone, physiological functions and action mechanisms of 6-DA. 

3. Materials and methods:

Comment: Line 122-127, the determination of APX is missing.

Reply: Lines 130-132, page 7 in the revised manuscript, we have supplemented the
determination method of APX activity.

4. Results:

Comment: Table 3 and Fig.1 can be merged, and the data of Table 3 should be added
into Fig.1.

Reply: We have mergeded Table 3 and Fig.1, and added the data of Table 3 into
Fig.1.

Comment: Line 276, APX is missing.

Reply: Line 279-280, page 14 in the revised manuscript, we have supplemented the “APX
(D)”.

Comment: Line 327, NAD-GDH (C), NADH-GDH (D).

Reply: Line 332, page 17 in the revised manuscript, we have supplemented the
“NADH-GDH (D)”.

5. Figures:

Comment: Fig 2-Fig 12, name of abscissa ‘Days after stress’ should be ‘Days after low
temperature stress’.

Reply: We have amended the name of abscissa by “Days after low temperature stress”
instead of “Days after stress”.

Comment: What’s the meaning of ‘b-d’ in the columns in Fig.2? Other Figures have the
same problem.

Reply: Statistical Analysis section, line 149-151, page 8 in the revised manuscript,
we have explained the meaning of letters in the columns in figures.

Comment: The letters in the columns of Fig.7 (A) is wrong, need to revise.

Reply: Fig.6 (A) in the revised manuscript, we have amended the letters error in the
columns of figure.

Response to Reviewer 2:

1. Comment: The abstract is poorly written in terms of language and clarity, please
rewrite it to give correct meaning, e.g L33. Mention the extent of decrease/increase
in studied parameters affected by LT or DA6. Which parameter was affected most?

Reply: Abstract section, Lines 26-39, page 2 in the revised manuscript, we have
rewritten the abstract in accordance with the comment, and mentioned the extent of
decrease/increase in the shoot and root fresh weight and dry weight affected by LT
or DA6, and addressed that the parameter (NiR activity) was affected most.

2. Comment: There are a lot of grammatical and spelling mistakes throughout the
manuscript, please rectify it and get it edited from native speaker, e.g. L44, L46,
L52, L57, L72, L87, L366, 396, 397.

Reply: We have polished the manuscript to aviod grammatical and spelling
mistakes.

3. Comment: L61,62, Please describe in detail about the possible role of DA-6 in
improving plant biochemical attributes, and which crops had already been tested for
this growth regulator. 

Reply: Introduction section, Lines 62-71, page 3 and 4 in the revised manuscript, we
have described the possible role of DA-6 in improving plant biochemical attributes
detailly, and the crops had already been tested for DA-6.

4. Comment: L78, there is no information about the study design (RCBD, CRD)? How many
replications/plants per container? How many plants were sampled at each sampling
time 0, 1, 3, 5, 7 days of treatment? Please clearly state when the plants were
sampled for growth parameters and Lp (only once or at each sampling time)?

Reply: Materials and methods, pages 5 and 8 in the revised manuscript, we have
addressed “the study design (RCBD)” in Lines146-147, “the amount of plants per
container” in Lines 95-96, “the number of plants were sampled at each sampling time
0, 1, 3, 5, 7 days of treatment” in Lines 99-100, and clearlied “the sampling time
for growth parameters and Lp” in Lines 97-98.

5. Comment: L107, at which stage the LiCor measurements were done? Which leaf was
sampled? If the leaf area corrections were performed relative to chamber (IRGA)
area? Why not the plants were analyzed/treated at 4 leaf stage? Justify please!

Reply: Materials and methods, page 6 in the revised manuscript, we have addressed
“the stage of the LiCor measurements were done” in Line 117, “leaf was sampled” in
Lines 119 and 120. The leaf area corrections were performed relative to chamber
(IRGA) area.

The reason for the plants were treated at 3 leaf stage and measured at 3～4 leaf
stage:

According to the result of our preliminary experiment, seedlings treated with DA-6
and/or LT at 3 leaf stage and measured at 3～4 leaf stage (0, 1, 3, 5, 7 days of
treatment in this study) showed large data variation range than seedlings treated at
4 leaf stage. Seedling at 3 leaf stage were treated could be favourable to observe
the data differences among the treatments, and futher analysis of DA-6 effect on
maize seedlings.

6. Comment: L112, 118, 122, 128… Please merge all these headings into a single
heading with appropriate terminology for mentioned biochemical analysis.

Reply: Materials and methods, Lines 115-116, page 6 in the revised manuscript, we
have merge the headings into a single heading.

7. Comment: L146, Please give more details about the design/factors for statistical
analysis, and which design was used for differentiation of treatment means? It is
encouraged to use Tukey test instead of LSD. How many experimental replications were
considered during statistical analysis? Why not the Data were analyzed by software
other than SPSS, as SPSS is generally used for social sciences?? Justify!!

Reply: Statistical Analysis section, Lines 146-151, page 8 in the revised manuscript,
we have detailed the design for statistical analysis, and number of experimental
replications. The results were statistically analyzed by the Tukey test in the
revised manuscript.

The reasons for SPSS as software in this study:

1) SPSS is one of the most powerful data analysis softwares, it is generally used for
social sciences, 

and is widely applied in crop chemical regulation research;

2) SPSS supports Chinese interface, could be easily-operated by non-statistical
professionals;

3) The function of SPSS meets the needs of this experiment.

8. Comment: L159, ‘on the 7th day.’ 7th day from what??? Mention the leaf stage in
parenthesis as well.

Reply: Results section, Line 165, page 9 in the revised manuscript, we have addressed
the starting point of the “7th day”, and mentioned the leaf stage in
parenthesis.

9. Comment: L363-365, It will be better to discuss results in purview of same crop –
maize, instead of rice or tomato. Also quote maize experimental findings for DA-6
effects, if available in literature. Reply: Discussion section, Lines 367-370, Page
19 in the revised manuscript, we have discussed the results in purview of maize,
instead of rice or tomato, and quoted maize experimental findings for DA-6
effects.

10. Comment: L390, Provide appropriate reference for the statement. 

Reply: Discussion section, lines 393, Page 20 in the revised manuscript, we have
provided reference for the statement.

11. Comment: Please update the references in discussion section, as most of the time
the references cited were from 2013-14.

Reply: Discussion section, we have updated the references.

12. Comment: Fig 1. It is strongly recommended to add at least 3 sets of roots
(replication) for each treatment, for the graphical representation.

Reply: Figs section, Fig 1 in the revised manuscript, we have added 3 sets of roots
for each treatment.

13. Comment: Fig 2. Please clarify, if the LSD test was performed using SPSS? It is
suggested to differentiate the treatment means only across the treatments, not for
the sampling intervals, because too much lettering does not give clear
understanding. Same is suggested for rest of the figures.

Reply: Figs section, we have clarified the Tukey test was performed using SPSS, and
differentiate the treatment means only across the treatments.

I hope the revision is satisfactory and please contact me shall you have further
questions.

Sincerely,

Jianguo Zhang

to Reviewers.doc
---

## [Decision Letter · Decision Letter 1]

13 Apr 2020

Exogenous Diethyl Aminoethyl Hexanoate Ameliorates Low Temperature Stress by
Improving Nitrogen Metabolism in Maize Seedlings

PONE-D-19-36030R1

Dear Dr. Wang,

We are pleased to inform you that your manuscript has been judged scientifically
suitable for publication and will be formally accepted for publication once it
complies with all outstanding technical requirements.

With kind regards,

Saddam Hussain

Academic Editor

PLOS ONE

Additional Editor Comments (optional):

Consider the following suggestions/changes

All the tables/figures should be self-explanatory. Avoid abbreviations in table
title. Define all the abbreviations used in the table foot note.

The following paper might be helpful in discussion

https://doi.org/10.3389/fpls.2018.00393

Conclusion should stand alone. At least avoid starting a sentence with an
abbreviation.

L175 in not clear.

Reviewers' comments:

Reviewer's Responses to Questions

**Comments to the Author**

1. If the authors have adequately addressed your comments raised in a previous round
of review and you feel that this manuscript is now acceptable for publication, you
may indicate that here to bypass the “Comments to the Author” section, enter your
conflict of interest statement in the “Confidential to Editor” section, and submit
your "Accept" recommendation.

Reviewer #1: All comments have been addressed

Reviewer #2: All comments have been addressed

2. Is the manuscript technically sound, and do the data
support the conclusions?

Reviewer #1: Yes

Reviewer #2: Yes

3. Has the statistical analysis been performed
appropriately and rigorously? 

Reviewer #1: Yes

Reviewer #2: Yes

4. Have the authors made all data underlying the
findings in their manuscript fully available?

Reviewer #1: Yes

Reviewer #2: Yes

5. Is the manuscript presented in an intelligible
fashion and written in standard English?

Reviewer #1: Yes

Reviewer #2: Yes

6. Review Comments to the Author

Reviewer #1: I am satisfied with the revision, but L174-176 in revision not describe
accurately about different lowercase.

Reviewer #2: Now the manuscript has been improved in the way suggested.

.

7. PLOS authors have the option to publish the peer
review history of their article (what does this mean?). If published, this will
include your full peer review and any attached files.

If you choose “no”, your identity will remain anonymous but your review may still be
made public.

**Do you want your identity to be public for this peer review?** For
information about this choice, including consent withdrawal, please see our
Privacy Policy.

Reviewer #1: Yes: Wang Yang Henan Agricultural University

Reviewer #2: No

---

## [Editor Report · Acceptance letter]

17 Apr 2020

PONE-D-19-36030R1 

Exogenous Diethyl Aminoethyl Hexanoate Ameliorates Low Temperature Stress by
Improving Nitrogen Metabolism in Maize Seedlings 

Dear Dr. Wang:

I am pleased to inform you that your manuscript has been deemed suitable for
publication in PLOS ONE. Congratulations! Your manuscript is now with our production
department. 

With kind regards,

on behalf of

Dr. Saddam Hussain 

Academic Editor

PLOS ONE